# Developing a high-quality patient-centric integrated model for emergency care system in selected districts of India: An implementation research protocol (INDIA-EMS Study)

Manu Ayyan S[1], Arvind Kumar Singh[2], Hemantkumar S. Patadia[3], Shreyas Patel[4], Saurabh Saigal[5], Jeyaraj Durai Pandian[6], Thejus Varghese[7], Ankur Joshi[8], Meenakshi Sharma[9]*, India-EMS Group Authorship[¶]

1 Department of Emergency Medicine & Trauma, Jawaharlal Institute of Post Graduate Medical Education and Research (JIPMER), Puducherry, India, 2 Department of Community & Family Medicine, All India Institute of Medical Sciences (AIIMS), Bhubaneswar, Odisha, India, 3 Faculty of Medicine, Parul Institute of Paramedical and Health Sciences, Faculty of Medicine, Parul University, Vadodara, Gujarat, India, 4 Department of Emergency Medicine, Parul Institute of Medical Sciences & Research, Parul University, Vadodara, Gujarat, India, 5 Department of Anesthesiology & Critical Care, All India Institute of Medical Sciences, Bhopal, Madhya Pradesh, India, 6 Department of Neurology, Christian Medical College, Ludhiana, Punjab, India, 7 Department of Emergency Medicine, Christian Medical College, Ludhiana, Punjab, India, 8 Department of Community and Family Medicine, All India Institute of Medical Sciences (AIIMS), Bhopal, Madhya Pradesh, India, 9 Division of NCD, Indian Council of Medical Research, New Delhi, India

¶ Membership of the India-EMS Group Authorship is listed in the Acknowledgments.
* smeenakshi@hotmail.com

## Abstract

### Introduction

Integrated emergency care systems are essential for achieving universal health coverage and managing time-sensitive conditions. In India, emergency care remains fragmented, with limited resources and coordination across healthcare tiers. The INDIA-EMS study aims to develop and evaluate a patient-centric, high-quality integrated emergency care model in diverse Indian districts.

### Methodology

The proposed implementation research is designed as a mixed methods study to be conducted over a 3-year period in selected districts of the states of Punjab, Gujarat, Madhya Pradesh, Odisha and Puducherry. The public/private medical college hospital/ tertiary care hospital in the district will serve as a hub and primary health care services as spokes. The health facilities will be graded according to the availability of resources for their readiness and preparedness to provide care for handling a particular emergency condition. The steps to build and implement the model are: 1. a gap analysis related to emergency care at both pre-hospital and health facility level; 2. a community-based survey in a sample

**Data availability statement:** No datasets were generated or analysed during the current study. All relevant data from this study will be made available upon study completion.

**Funding:** This study protocol INDI-EMS/V9 version was technically approved by the funding agency and the study will be supported by funding from the Indian Council of Medical Research (ICMR), India under NHRP file number. NHRP-2023-0000013.The funder was involved in identifying the research priority, protocol development, and review of the manuscript. The funder will also continue to monitor the study until completion.

**Competing interests:** The funder has appointed the study monitor MS [Scientist G ,Division of NCD , ICMR, New Delhi], under whose guidance the research will be conducted. The funder will not be involved in data collection, analysis and interpretation of the results. The study monitor and coordinator will however oversee that the timelines are met and study is being conducted as per protocol and following the ethical standards. This does not alter authors' adherence to PLOS ONE policies on sharing data and materials.

**Abbreviations:** CFIR, Consolidated Framework for Implementation Research; CHC, Community Health Centres; DH, District Hospital; ECS, Emergency Care Systems; EDB, Emergency Disease Burden; EDMR, Emergency Disease Mortality Rate; EMD, Emergency Medical Diseases; GOI, Government of India; MCH, Medical College Hospital; MLC, Medico Legal Case; MoHFW, Ministry of Health and Family Welfare; NITI Aayog, National Institution for Transforming India; NP-NCD, National Program for Non-Communicable Diseases; PHC, Primary Health Care Centres; PMJAY, Pradhan Mantri Jan Arogya Yojana; PSU, Primary Sampling Unit; RKS, Rogi Kalyan Samiti (Patient Welfare Committee)/ Hospital Management Society; SDH, Sub-District Hospital; STW, Standard Treatment Workflow; TAEI, Tamil Nadu Accident and Emergency Care Initiative.

of 30,000 in 5 districts for EMD burden estimates and health-seeking behaviour at baseline and endline; 3. use a consolidated framework for implementation research to develop, optimise and implement hub and spoke model through three cycles of iterative processes; and; 4.evaluation for feasibility, acceptability, cost, effectiveness and coverage.

## Discussion

A high-quality patient-centric integrated emergency care model may be able to ensure efficient delivery of care to patients experiencing time-sensitive emergencies and advance towards the coveted target under Sustainable Development Goals (SDGs).

## Ethics and dissemination

Ethics approval was obtained in all the project sites. The results of the project will be submitted to a peer-reviewed journal for publication, in addition to national and state-level dissemination.

## Trial registration

The trial is registered with CTRI (Clinical Trial Registry of India; CTRI/2024/01/061304).

---

## Introduction

India is committed to achieving Sustainable Development Goal 3 by 2030, which aims to "ensure healthy lives and promote well-being for all at all ages," wherein emergency care systems are pivotal for addressing universal health coverage, road safety, maternal and child health, and various disease burdens. The World Health Organization (WHO) underscores that the absence of timely emergency care significantly contributes to public health crises, with the 72nd World Health Assembly highlighting timeliness as a critical dimension of care quality [1]. The World Bank Disease Control Priorities Project estimates that a high-quality integrated emergency care system could address over half of deaths and a third of disabilities in low- and middle-income countries (LMICs) [2].

Population-level incidence data for emergency medical diseases (EMDs) are scarce; consequently, mortality data are predominantly used to estimate their burden. Globally, emergency EMDs contribute significantly to health burdens, accounting for approximately 28.3 million deaths annually and disproportionately impacting low-income countries. South-East Asia bears a notably high burden, with 90% of deaths and 84% of disability-adjusted life years (DALYs) attributable to emergency conditions. Improved emergency care systems can notably reduce mortality from cardiovascular diseases, neonatal conditions, and maternal conditions. Ischemic heart disease, stroke, neonatal conditions, lower respiratory

infections, and a combined category of injuries and road traffic accidents were five leading EMD causes of death in India in 2019 [3]. The combined EDMR of these EMDs was 251 deaths per 100 000 population (111, 51, 32, 29 and 28.3/100,000 population respectively) in all age groups [4,5]. These EMDS were thus responsible for 37.4% of all the deaths in the country. The country also has the highest absolute number of deaths (51,000) due to snakebite in the world (4/100,000 population) [6]. India with a maternal mortality rate of 6/100,000 in the age group of 15–45 years accounts for 12% of the global deaths [7,8]. The leading cause of these maternal deaths were obstetrics haemorrhage (47%) and hypertensive disorders of pregnancy-eclampsia and preeclampsia (7%) in 2019 [9,10]. In view of this, we plan to target time-sensitive emergencies STEMI, stroke, trauma and burns, acute respiratory illness, post-partum haemorrhage and preeclampsia, neonatal emergencies, snake bite and poisoning for building an integrated emergency care model.

India's emergency care system, while evolving, confronts several deeply entrenched challenges. The Government of India's public policy think tank, NITI Aayog's 2021 country level assessment of emergency care delivery brought out various observations. A primary issue is its fragmented nature, leading to a disjointed care continuum from pre-hospital settings through primary, secondary, and tertiary facilities [3]. This fragmentation manifests in multiple uncoordinated initiatives, often establishing separate specialized centers (e.g., for trauma, burns, pediatrics) which can lead to resource duplication without proportional returns on investment. Pre-hospital care is rudimentary, with ambulance systems often suffering from poor infrastructure, a lack of trained personnel, inadequate access, and systemic fragmentation, despite the expansion of services like Dial 102/108 [3,11,12]. Pre-arrival notification to receiving hospitals is often lacking [3]. Further challenges include significant gaps in healthcare infrastructure, workforce shortages, and inequitable distribution of medical provisions, particularly outside tertiary centers [3]. Limited community access to timely care, poor coordination between health facilities, a scarcity of standardized training resources, and low public awareness regarding emergency responses exacerbate the problem. Most emergency services are concentrated at the tertiary level, leaving prehospital, primary, and secondary care systems with limited capabilities. Additionally, there is an absence of robust public-private partnerships, and many attempts to address these issues have been system-focused rather than patient-focused, failing to bridge critical gaps in care delivery and effectiveness.

To address these multifaceted gaps, there is an urgent imperative to develop and implement an integrated emergency care system model. Such a model would aim to manage time-sensitive emergencies more effectively by establishing a seamless continuum of services across different healthcare facilities, interconnected through robust communication, transportation, and referral systems. However, large-scale implementation research employing an integrated approach to develop a scalable, patient-centric emergency care system has not been done in India. Using an implementation research framework the project aims to understand the contextual factors that facilitate or hinder the transition to high-quality emergency systems, providing evidence-based guidance for policymakers and planners for resource allocation, program development, and scaling up successful models nationally and globally. This approach will move beyond system-focused fixes to create a truly patient-centric emergency response framework.

## Aim and objectives

The aim of this study is 'to develop a scalable model to achieve 80% population coverage with high-quality patient-centric integrated emergency care system in the selected districts of India' with specific objective to:

1. Co-develop a district-level implementation model for high-quality patient-centric integrated emergency care through iterative processes.

2. Evaluate the model in terms of feasibility, acceptability, cost, and effectiveness of the implementation to achieve emergency care preparedness and coverage.

## Materials and methods

### Study population and sites

Five districts (Ludhiana, Vadodara, Vidisha, Puri and Puducherry) from 5 Indian States/UTs (Punjab, Gujarat, Madhya Pradesh, Odisha and Puducherry) will be participating in this study. The districts are geographically diverse, have a varied health index, healthcare infrastructure, diverse emergency care challenges, incidence rates and innovative practices (Fig 1). The selected districts are a purposive sample of districts that attempts to be representative of the cross section while ensuring support from state health departments and access to research teams.

We plan to use a hub and spoke model for managing the eight EMDs. This is based on the model envisaged in the STEMI guidelines under the National Programme for Prevention and Control of Non-Communicable Diseases (NP-NCD) 2022 [13]. The medical college hospital/ tertiary care hospital in the district will serve as a Hub and the various public health facilities District Hospital (DH), Community Health Centres (CHCs) and primary health care centres (PHCs) as spokes. In order to provide care to these EMD patients to be managed in a time sensitive manner, we plan to include hospitals from the private sector too with infrastructure and capacity to handle one or all of these emergency conditions.

The NP-NCD 2022 guidelines for STEMI management also recommend that participating facilities may be graded according to the availability of resources. The following are the levels of facilities and the recommended package of services for the management of STEMI:

L-1: Facility having a medical officer with no ECG facility and not capable of thrombolysis.

L-2: Facility having a medical officer with ECG facility and capable of thrombolysis.

L-3: Facility with a medical officer, MD medicine, technician, nurse, and having emergency care (ICU/HDU) set up with thrombolysis capability.

L4: Hospitals where thrombolysis/PCI for STEMI is already happening.

We plan to use a similar method for grading the public and private health facilities for providing care for a particular emergency condition. Each facility may be at a different level for different conditions based on the readiness and preparedness of each facility to handle different EMDs. These classifications are likely to change during the implementation based on improvements in infrastructure and manpower and availability of new facilities.

### Study design and overview

We plan to conduct a multi-district implementation study using a mixed-methods design that combines formative and evaluation components through three iterative improvement cycles to develop and implement a high-quality emergency care system. Our model will provide quality care to patients with STEMI, stroke, trauma and burns, acute respiratory illness, postpartum haemorrhage and preeclampsia, neonatal emergencies, snake bite and poisoning. The study will be of a three year duration. It will be conducted in collaboration with the state health department.

The study will start with (1) Identification of gaps related to emergency care at pre-hospital, public and private healthcare system; (2) Simultaneously, a community based survey will be done to obtain baseline data on load of EMDs and level of community awareness; (3) Formative research will be done to co-develop a district specific provisional patient centric integrated model of emergency care; (4) optimization of the model through three cycles of iterative processes to provide continuum of care across different levels of the health system; 5. Implementation of the model through hub (tertiary care hospitals) and spokes (a cluster of public health facilities- DH, CHC, PHC) linked via pre-hospital system in a designated geographical area in the district and 6. Evaluation of the model for feasibility, acceptability, cost, effectiveness and coverage in the above five selected districts. It will then be scaled up to the entire district in consultation with the respective state governments and its impact on population coverage will be evaluated (Fig 2). The study results will be disseminated to policy makers at state and national level.

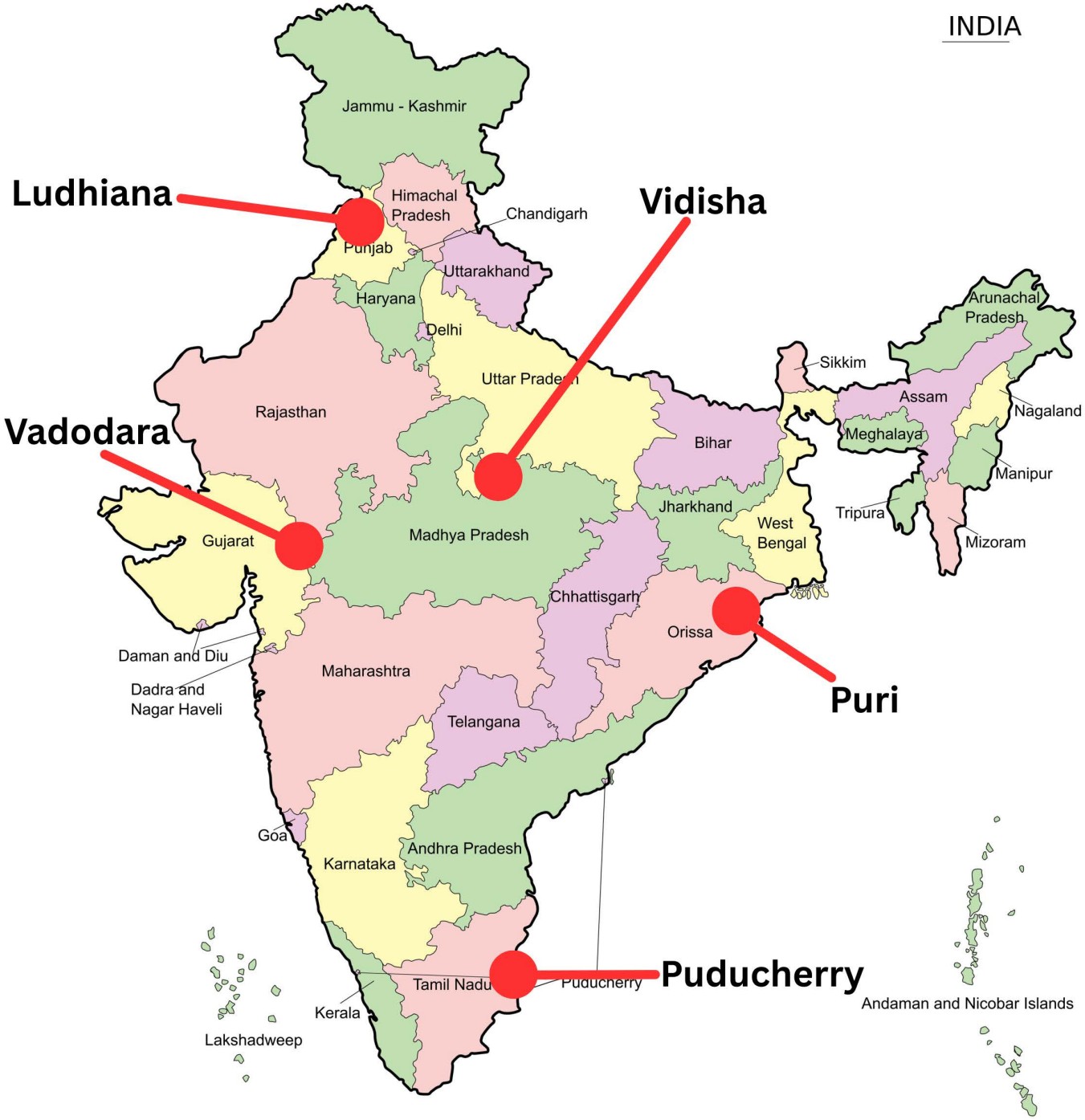

**Fig 1. India-EMS Site Map.** Reprinted from https://commons.wikimedia.org/wiki/File:India-map-en.svg#filelinks under a CC BY license, with permission from Rajesh Odayanchal, original copyright 2011.

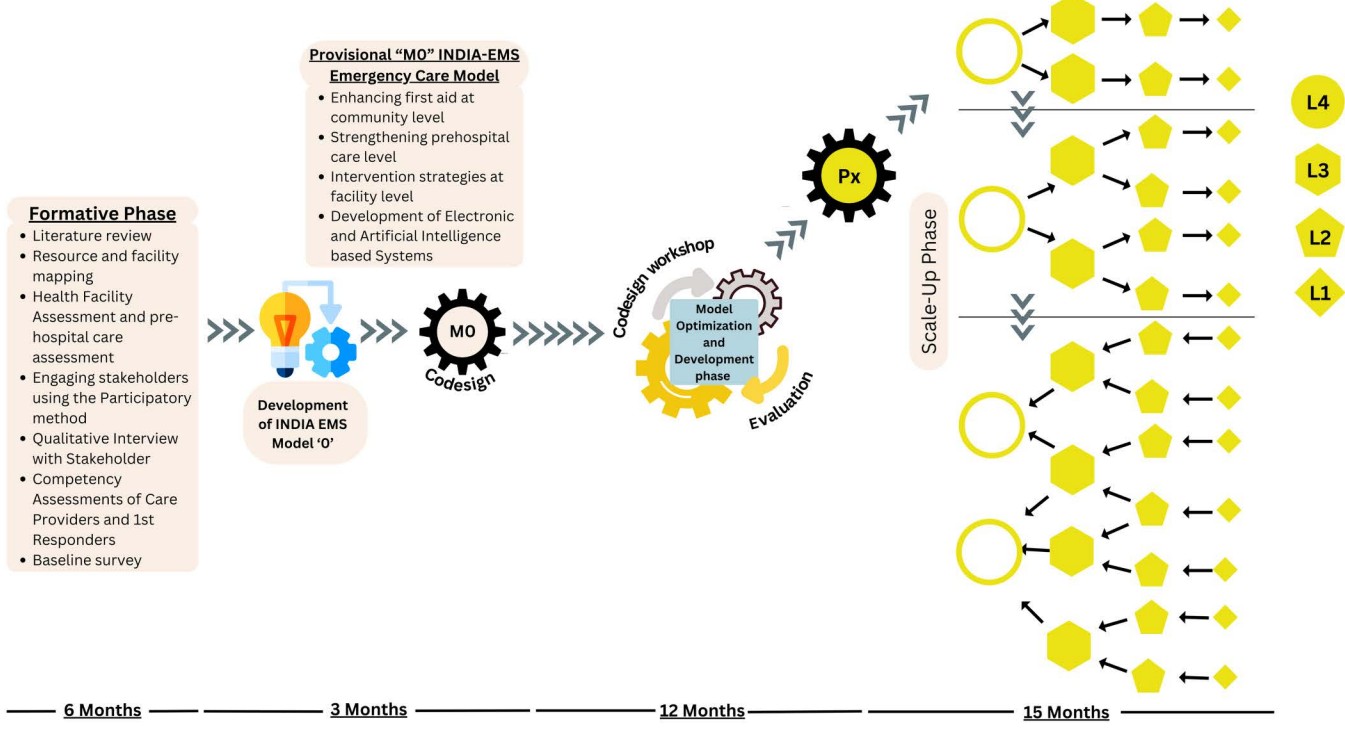

**Fig 2. Overview of the study.**

## Study stakeholders

(a) Patients who have experienced an emergency condition within the study geography during the study period and have visited a health facility,(b) Community or households, caregiver of patients,(c) Local political leaders (including Pancha-yats and Block Development Officers), village self-help groups, RWAs, civil society members,(d)Potential first respond-ers: Road safety authority (including traffic police), ambulance drivers and paramedics community groups, taxi and auto drivers, shopkeepers, bystanders, family members etc.,(e) Health care providers and managers: Staff and administra-tion of health facilities, including both public and private sector,(f) Health policy makers and health system management involved in the management and decision-making processes of the emergency care system, e.g., Secretary Health, Mis-sion Director- National Health Mission, Directorate of Health Services, public health and medical education, nodal officer trauma and emergency care, any other key personnel as identified by the State), National Health Authority, regulators, and legal advisors.

## Study team

The study team consists of (a) the research team and (b) the implementation team. These teams will co-develop the district-level emergency care model.

*Research team*: This comprises researchers, state and national health department officials and ICMR. The research team will be primarily responsible for formative research, supporting the implementation of the intervention, and perform-ing data collection for iterative program learning and quality improvement, and assessment of the outcomes. The research team is further divided into three individual teams i) the formative research and program learning team; ii) the implementa-tion support team; iii) the outcome measurement team.

*Implementation team*: This comprises policymakers and administrators at district, state, national level governments, public health care providers and managers, private healthcare facilities and providers, ambulance providers, non-governmental organisations, civil society, non-health departments like transport, education, and communication. The implementation team will be primarily responsible for implementation of the provisional and final model at the district level.

The research team and implementation team will co-develop district level emergency care model. Key roles, staff involved, and key activities of the study team are outlined in Table 1

## Formative research

The objective of this phase is to conduct a situational analysis to understand the current state of emergency care systems in the study geography, with the aim of identifying the factors that act as potential barriers and facilitators. This will help in identifying the means to transform the existing system to a high-quality integrated emergency care system.

Though integrated emergency care systems are considered a solution for providing accessibility to emergency care in diverse geographical settings, evidence based information for planning and implementing a patient centric integrated emergency care system, particularly in low resource settings, is required. Therefore, in the initial phase of the study we plan to conduct:(i) a systematic literature review: to synthesise evidence for an integrated emergency care system, which has been adopted globally or within India to meet the needs of the population served. This will provide an overview of what has been tried to be integrated and its purpose; how this can be used to develop the model, processes, strategies and structure within the context of the needs of the population of selected districts. (ii) health system readiness: a. health facility and pre-hospital (ambulance and helpline centres) resource and infrastructure assessment in the study district and

**Table 1. Description of study teams.**

| Teams | Implementation Team (Government and Private partners) | Research Team (Principal investigators, co-investigators, state health officials, research staff recruited under the study, scientist at ICMR, project steering group and technical advisory group) | | |
|---|---|---|---|---|
| Overall role | Co- develop the provisional model Implementation of the provisional and final model | Conduct formative research, co-develop the intervention, support the implementation of the intervention, perform data collection for iterative program learning and quality improvement, and assessment of the outcomes | | |
| Staff/ Stakeholders | Policymakers and administrators in the state governments, district health administration, health care providers and managers at district level, private healthcare facilities and providers, ambulance providers, non-governmental organizations, civil society, non-health departments like transport mass media communication and education. | **Formative research and programme learning team** | **Implementation support team** | **Outcome measurement team** |
| Activities | • Providing overall leadership and governance.<br>• Co-develop the provisional and final model.<br>• Create an enabling environment and provide managerial and administrative support for implementation, scale up, and institutionalizationon of the interventions.<br>• Implement activities related to the provisional and final model.<br>• Provide resources required for effective implementation of the model | • Perform formative research to co-develop the provisional model.<br>• Qualitative data collection at regular intervals for program learning, reiterations, and continuous quality improvement.<br>• Contribute to refinement and optimization of the model | • Support the implementation team in health facility preparedness for providing high-quality emergency care.<br>• Contribute to refinement and optimization of the model<br>• Provide support to the implementation team to implement the provisional models and the final model. | • This team will be independent from the formative research team and the implementation support team.<br>• Quantitative data collection for concurrent and end evaluation for emergency care preparedness and coverage.<br>• Contribute to refinement and optimization of the model |

a defined area in adjoining districts; b. Mapping of emergency care system: geo spatial mapping of facilities; mapping of current emergency care pathways, care processes and referral pathways using observation checklists, direct observation, and interview with health care providers; c. competency assessment of care providers and 1st responders to assess the knowledge and skills of healthcare providers; d. budgetary provisions abstracted from administrative data including public and private insurance systems and; e. indicators of the current state of emergency care, e.g., hospital emergency visits, types of emergencies, patient outcomes.(iii) A community-based household survey: to estimate the burden of EMDs in the community and assess the health care services sought after the event. The survey data will provide information on the proportion who sought care, pathway of care & referral pathway, delay in seeking care, the expenditure incurred during care provision for the emergency medical condition and utilisation of insurance services. A multistage sampling strategy will be adopted. The study population will be stratified into urban and rural areas. Villages will be the primary sampling unit (PSU) in rural areas. Villages with a population of less than 500 will be excluded from the sampling frame, and villages with a population of more than 5000 will be split into clusters to have an approximate population of around 2000 in each cluster. In an urban area, census enumeration blocks will be a PSU. Sample size is calculated to estimate the burden of emergency medical conditions. Considering the burden of emergency conditions as 141/10000 population (Table 2), assuming a confidence level of 95% and a relative precision of 25% and design effect of one (design effect is considered as one as due to rarity of outcome the events are less likely to be clustered), the required sample size is 4292. Considering a non-response rate of 20% the sample size comes out to be 5365 which is further rounded off to 6000. A total of 60 PSUs will be randomly selected (PSU stratification for urban and rural will be based on district's urban:rural distribution) and 100 households will be randomly selected from each PSU. All the individuals in the selected household will be included for the purpose of data collection.

The qualitative component of the formative research will involve semi-structured interviews, focus groups, and surveys with key stakeholders, including administrators, policymakers, healthcare providers, first responders, paramedics, community groups, and patients. These interactions aim to gather diverse perspectives on state and district-level policymaking, program implementation, and financing options. Additionally, experiences and best practices from existing state-led programs like TAEI in Tamil Nadu and ambulance systems in Gujarat and STEMI and stroke models in Ludhiana will be explored [14–16]. Components of formative research, stakeholders, data collection techniques and sample size for each activity is provided in S1 Table.

**Table 2. Burden of emergency conditions.**

| Baseline coverage (%) | End line coverage (%) | Number required who had an emergency medical condition | Population to be surveyed |
|---|---|---|---|
| 10 | 20 | 216 | 16241 |
| | 30 | 69 | 5188 |
| | 40 | 36 | 2707 |
| 20 | 30 | 311 | 23383 |
| | 40 | 89 | 6692 |
| | 50 | 43 | 3233 |
| 30 | 40 | 373 | 28045 |
| | 50 | 101 | 7594 |
| | 60 | 46 | 3759 |
| 40 | 50 | 405 | 30451 |
| | 60 | 104 | 7820 |
| | 70 | 46 | 3459 |
| | 80 | 25 | 1880 |

## Development of provisional model '0'

A conceptual framework will be co-developed to frame the intervention framework involving the key stakeholders based on review, formative research and situational analysis. A co-design workshop will be conducted in each district under the leadership of the state government, attended by the research team, district program officers, community leaders, hospital managers, doctors, nurses, and paramedics from the participating hospitals and officials from non-health departments including transport, education and communication. The model will be co-developed based on the emergency care system framework and health system building block, encompassing community, prehospital and hospital care under various health system building blocks (e.g., infrastructure, equipment, supplies and provisions, financing, governance and leadership, information systems). Visual representation showing the description, relationships, and interactions with the different components in the provision of emergency care will be identified to co-develop the model. System-level and emergency medical disease-specific interventions will be developed and fine-tuned based on the workflows developed by the research team.

The core components of the provisional model will include (a) enhancing first aid at the community level, (b) strengthening prehospital care; (c) intervention strategies at the facility level; and (d) electronic and artificial intelligence-based decision support systems (Fig 3).

## Model optimization and development phase

Pilot testing of the model (M0) will be done in a limited sample and geographical setting which is likely to be representative of the entire district in consultation with the state. A single hub and spoke cluster will be chosen to have a representation

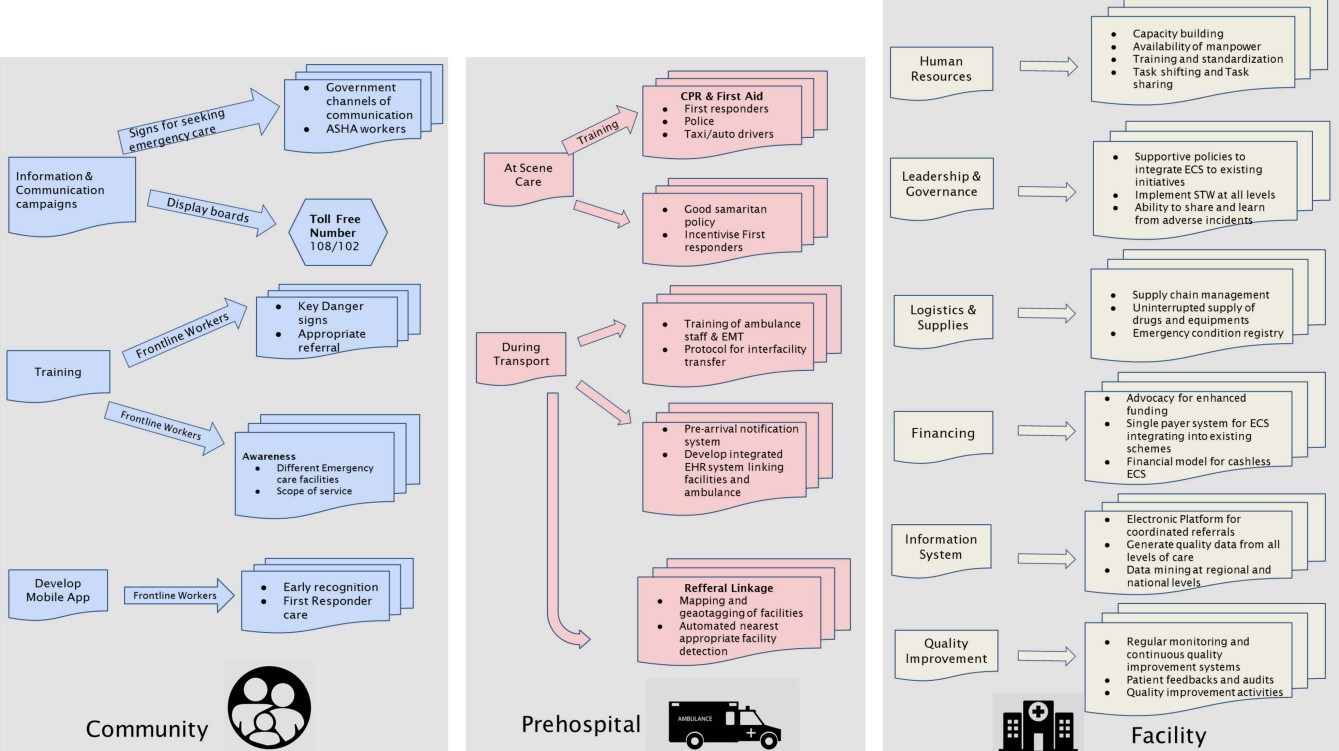

**Fig 3. Provisional model emergency care framework for co-development of Model '0'.**

of each level of health facility (prehospital care, Level 1, Level 2, Level 3, and Level 4) for each of the emergency conditions. Pre- and post-implementation data will be analysed to assess acceptability, feasibility and effectiveness based on process and quality outcome measures. Quantitative data on key indicators on the implementation strategy (e.g., effectiveness of training of care providers, appropriateness and effectiveness of educational materials), process indicators (e.g., adherence to new care protocols, patient satisfaction), and implementation measures (e.g., acceptability by the care providers) will be collected. The model will be refined based on a review of the provisional model, data collection and evaluation of the key indicators. As described in the formative research multi-stakeholder consultation (Co-design workshop), in-depth interviews and focus group discussions will be done to get further insights. Key areas will be identified where the provisional model has not worked effectively. Necessary adjustments will be made to the model in a consultative manner. Subsequent cycles will be developed by re-iterating the same process. A total of 3 iterative processes will be done to refine the model and to develop the final model (Px) (Fig 4). Each disease process will be deployed sequentially, and iterative processes and periods will be conducted accordingly. The threshold for initiating the scale-up phase will be decided during the formative research phase after the baseline surveys and mapping by the research team. The threshold will be multidimensional and the irrespective of attaining the threshold after 3 iterative cycles scale up will be initiated.

### Implementation framework to guide the development and optimization of the model

We will use the consolidated framework for implementation research (CFIR) to examine complex health interventions, combining theoretical constructs and an active change process that is adaptable for emergency care systems (ECS) (Fig 5). It ensures that interventions are contextually relevant, addressing various facets of ECS through five distinct

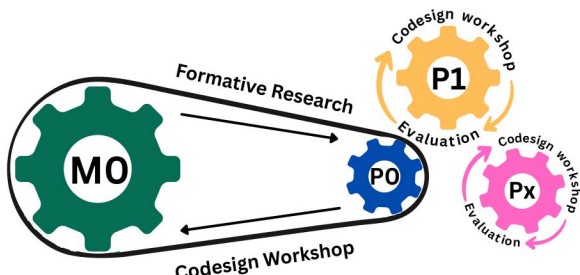

**Fig 4. Iterative model optimisation process.**

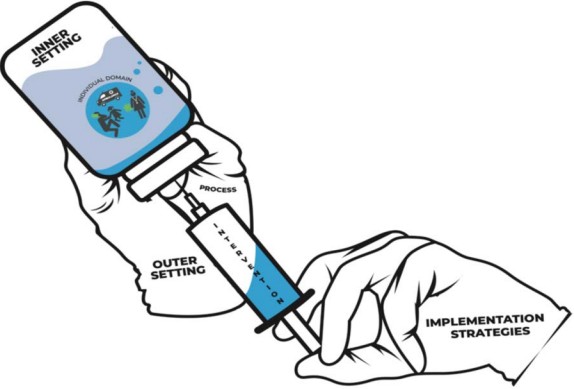

**Fig 5. CFIR framework used for INDIA-EMS model.**

domains: outer settings, inner setting, individuals involved, intervention characteristics, and the process of implementation. The determinant framework used for implementing high quality ECS in this study is described in S2 Table. Within these domains lie 48 constructs, each capable of acting as a barrier or facilitator to implementation [15,17]. The flexibility and inclusiveness of CFIR makes it suitable for participatory action research, systematically assessing elements influencing both implementation and effectiveness.

### Scaling up of the final model

The piloted final model 'Px' will be scaled up step-wise to the entire district in a phased manner. The scaling-up sequence will start from the hub and spoke system where pilot testing was conducted and then progressively moved to the other spokes and other hubs of the district. A detailed scale-up strategy, tools to be used for scale-up and scalability of the model will be co-developed by consultation with the state. The objective during this phase is to implement the final high-quality integrated INDIA-EMS model across the whole district to achieve 80% population coverage of this model. This rollout will be done primarily by the district administration (the implementation team), with support from the program learning team. The outcome assessment team will conduct the concurrent evaluation and the end-line population survey.

### Evaluation of the India-EMS model

The primary aim of the study is to develop an optimized model of integrated high-quality emergency care, which will achieve a population coverage of 80% in the district.

**Primary outcomes.** The primary outcome indicator is coverage (S3 Table), i.e., the proportion of patients who were provided definitive care for the specified emergency medical condition at an appropriate health facility in a time-sensitive manner. The appropriate facility will be defined during the facility mapping for each emergency condition. Time sensitivity will be defined as per disease condition and decided based on expert consultation and available evidence during the formative phase.

I. *Emergency health care coverage indicators that would be covered are:*

 a) Proportion of patients who sought care for any emergency medical condition

 b) Proportion of patients who used ambulance services to reach health facility for a emergency medical condition

 c) Proportion of times an ambulance responded to the emergency calls.

 d) Proportion of patients who were provided definitive care for the specified emergency medical condition

 e) Proportion of patients who utilized PMJAY/state government insurance schemes during care for emergency medical conditions

 f) Proportion of patients who had catastrophic expenditure during care for emergency medical conditions

The coverage indicator will be disaggregated for different emergency conditions
The second indicator which will be assessed is preparedness of the emergency health care.

II. *Emergency health care preparedness indicators*

 a) Proportion of health facilities having a fully equipped functional ambulance

 b) Proportion of ambulance personnel trained in "at scene and transit" management of common emergencies

 c) Proportion of lay first responders trained in "at scene" management of common emergencies

d) Proportion of health facilities having round-the-clock availability of complete emergency care team trained in management of medical emergencies/trauma/poisoning

e) Percentage health facilities having availability of key/essential equipment and medications required to provide definitive care for specified emergency condition

f) Percentage health facilities having an infrastructure readiness score above a desirable level

g) Proportion of health facilities providing care as per Standard Treatment Workflow (STW) and Standard Operating Procedure (SOP) as per the level of care for the management of emergency medical condition

h) Proportion of health facilities conducting regular mortality audits of emergency medical condition

i) Proportion of health facilities utilizing RKS funds for emergency care

**Secondary outcomes.** The secondary outcome of the study will be Implementation outcome and process outcome indicators.

*Implementation outcome:* Implementation of the model will be assessed for following three parameters; acceptability (patient satisfaction and acceptability of health care workers), cost (cost incurred by the health system and project team in implementation of the model), and feasibility (challenges faced during the implementation, adoption of the model and program learning). During the scaling-up phase following implementation outcomes will be captured.

*Process outcome indicators:* These indicators will be used during formative research for optimising the model (S4 Table). These will feed into the codesign workshops, which will help refine the model every 3 months and lead on to building the final Px Model. They include time to arrival and time to treatment for selected emergency conditions. These will be measured by the program support team every three months for 24 hours by live observations. They will also be modified and adapted during the formative research phase.

Evaluation will be done at baseline, concurrently during the implementation of the model and at the endline(S5 Table). Baseline, concurrent evaluations, and end-line survey data will be analysed to assess the feasibility, acceptability and the cost of the implementation of the model, and the effectiveness of the implemented model in terms of emergency care preparedness and coverage. Concurrent evaluation: Health facility assessment, qualitative interview with stakeholders, competency assessment of care providers will be done concurrently at an interval of three months. Post-care interviews (telephonically) of patients will be done to assess coverage and timeliness of care, expenditure, pathway of care, and delay in care. Endline evaluation: Endline survey will be conducted after the completion of the implementation phase. The sampling strategy and methodology of the endline survey will be similar to the baseline survey. Objective of the endline survey will be to estimate the increase in population coverage and health facility preparedness as compared to the baseline. The estimate of population coverage (i.e., reaching the appropriate health facility within a specified time frame as per the condition; this time frame will be decided during the formative research) obtained during the baseline survey will be used to calculate the sample size for the endline survey. Thus, the sample size for the endline survey will be calculated after the analysis of the results of the baseline survey. We are providing an illustration for sample size required for different scenarios of population coverage assuming a power of 80%, the confidence level of 95%, and a non-response rate of 20%. Thus, sample size included in baseline survey (at the district level) will be sufficient if there is at least 20% increase in population coverage from pre-intervention to post-intervention.

## Data collection instruments

Community survey and health care provider/stakeholder survey will be done using a structured questionnaire to capture the occurrence of an emergency medical condition in the last 12 months and treatment seeking behaviour, and barriers and facilitators in seeking care. We will also capture the electronic health record data for disease burden, care and

outcome related information. Competency of health care providers will be assessed using knowledge and skill observation checklist. Gap analysis of the health facilities will be done using a structured questionnaire which will be developed during formative research based on review of literature and expert consultation.

## Data management

Data management will involve the use of a paperless data collection tool and will include geo-spatial tagging and analysis (using space-time matrix and kernel density), development of a Health Management Information System (HMIS) which includes registers, reports, and databases, real-time supportive supervision, interim analysis for process indicators through serial analysis, internal checks for data validation, text mining for qualitative data, and an iterative "inductive" framework analysis. The digital tools and the platform will be developed along with the ICMR HQ and the Gujarat team. The data servers and management will be hosted at ICMR. The qualitative data from FDG, IDI's and semi-structured interviews will be transcribed and translated at the study site and analysed. The quantitative data management will be done at the respective study site.

## Progress of the study

The study began on 12 January 2024 in all study sites. The participant recruitment is estimated to be completed by December 31, 2027, followed by data analysis and publication of results by February 2028.

## Data analysis plan

The collected data will be analyzed using both quantitative and qualitative methods

**Qualitative analysis.** We propose to adapt grounded theory in this research looking at the nature of inquiry. We will start with open coding and try to identify concepts in the data without predetermined categories. Through constant comparison, codes will be refined, and will be converted into categories. All the categories will be leading to the development of a theoretical framework. As this phenomenon is also nearer to applied policy research, we further propose to do a framework analysis where we will use a matrix to organize and interpret data by themes and categories. After the thematic analysis, we will adapt a semi-quantitative text mining approach where we will be converting text to a tokenized data frame and stop -words will be anti-joined to this date frame. The word frequencies will be plotted using bar graph and by word cloud diagram. Finally, we will explore the correlation between words and a network plot of the same will be plotted

**Quantitative analysis.** The data will be duly checked for duplication, redundancies, missing values, and outliers. The key variables will be identified and summarized by measure of central tendency and measure of dispersion as per variable type. The incidence of emergency medical conditions will be calculated separately for urban and rural areas and pooled weighted estimates will be calculated based on distribution of population in urban and rural areas. The effect of the implementation model will be assessed statistically using appropriate methods as described here; We will perform the repeated measure Analysis of variance (ANOVA) as a univariate analysis for those dependent variables which are continuous in nature and having a within-subject correlation. The multivariate analysis for the same variables will be performed by linear mixed effect models (to account for population level and individual level variability). We further propose to calculate a population-average effect by Generalized Estimating Equations (GEE) looking at the structure of the data collection plan and robustness of GEE in the case of model misspecification. The temporal pattern (if any) in some variables will be checked by autoregressive integrated moving average (ARIMA) and if any temporal pattern is indicated we will perform a time-to-event analysis using Cox proportional hazard model.

**Geo spatial analysis.** We propose to perform the space-Time Cube method which will allow us to explore the spatiotemporal pattern (if any) of changing the variable of interest in reference to space and time. In this three-dimensional

representation one axis will represent the spatial location, another will represent the time and the third axis will represent the attribute of interest. We will attempt to identify clusters, trends, outliers, or patterns by this method. We will also try to visualize the effect of different models by filtering data on the specific spatial and temporal criteria. We further propose to do a hotspot analysis to identify temporal variation and statistically significant clustering. These identified clusters will be checked for spatial proximity or similarity using k-means clustering. We will combine or overlay the different layers of data to check for any relationships or patterns visually and will perform point -in -polygon queries and intersection analysis. The degree of similarity between spatial features will be calculated by spatial autocorrelation statistics, such as Moran's I or Geary's C which will help to identify clustering or dispersion patterns.

**Data analysis software.** Statistical software STATA/R-programming environment will be used to manage and analyze the quantitative data. The transcribed data will be examined through thematic analysis utilizing NVIVO software (QSR International Pvt Ltd).

## Ethical considerations

This study will adhere to the ethical standards of the Good Clinical Practice Guidelines and the ICMR National Ethical Guidelines 2017. Ethical approval was received from the Christian Medical College and Hospital Institutional Ethics committee (Reference Number: BMHR-IEC/Apprvl-IndiaEMS/23-12-503/Neuro), Jawaharlal Institute of Postgraduate Medical Education and Research Institutional Ethics Committee for Observational Studies (Reference Number: JIP/IEC-OS/2023/352), Parul University Institutional Ethics Committee for Human Research (Reference Number: PUIECHR/PIMSR/00/081734/6401), All India Institute of Medical Sciences, Bhopal, Institutional Human Ethics Committee, (Reference Number: IHEC-LOP/2023/EL0116), All India Institute of Medical Sciences, Bhubaneshwar, Institutional Human Ethics Committee (Reference Number: T/EMF/CM&FM/2023–24/104) with additional permissions from respective district health authorities. Principal investigators at each site will manage study-related oversight. The study involves the implementation of standard care treatment workflows requiring no participant consent. However, blanket consent will be obtained from district health administration or facility managers in accessing registers and facility-level data. Prospective study participants will be inducted only upon obtaining written informed consent, with an additional provision for assent in the case of younger participants. All individual in-depth interviews and focus group discussions will be conducted in confidential settings. Any adverse events or protocol amendments will be promptly reported to the Institutional Ethics Committee, ensuring no protocol changes without written consent.

## Dissemination of the findings

A state-level and national-level workshop will be conducted after the implementation of study activities. The objective of the workshop will be to disseminate the study findings, inform the policymakers and decision-makers, and enable discussions for scaling up the implementation model at the national level.

## Project management and role of partners

The Ministry of Health and Family Welfare (MoHFW), Indian Council of Medical Research (ICMR), state governments, and investigator teams are the key partners in this project. MoHFW will provide leadership, technical support, and policy guidance. They will also organize training and meetings, and disseminate best practices. ICMR will support research, monitoring, evaluation, and upscaling of the project. They advocate for emergency care, develop protocols, conduct training, and review progress. State governments are responsible for implementing the project at the state level. This includes ensuring healthcare staff participation, access to equipment and drugs, ambulance availability, staff training, and project monitoring and evaluation. ICMR plays a key role in supporting and implementing the emergency care project. They advocate for emergency care, develop protocols for various aspects like drug supply and insurance, and conduct training and

meetings. They also visit other institutes and participate in meetings to share best practices and expertise. Additionally, they provide technical inputs and support for scaling up the project effectively. The project management organogram is shown in S1 Fig.

While the Project Selection Committee will help to develop the research protocol for the emergency care project, they will also support the project's expansion and ensure it meets targets. They provide technical expertise and resources to monitor progress and offer feedback, thus providing inputs to improve the project based on past or similar experiences. The Program Steering Group will provide overall leadership and guidance. Moreover, they will review research and project scale-up. They will be indulged in direct national and state actions. It will include representatives from various partner institutions. The Technical Advisory Group will oversee project implementation research and scale-up. They will review progress and ensure seamless integration into national programs. It includes leaders from stakeholder agencies.

## Discussion

Despite emergency services being envisioned as a unified point of entry for all emergencies, they face challenges due to fragmentation across various emergency conditions and levels of care. This fragmentation is evident in multiple initiatives across the country that establish separate centers for specific conditions such as trauma, burns, paediatrics emergencies and obstetrics emergencies with dedicated emergencies or casualties. While the rationale for establishing these specialized centers is often presented as to improve the quality of care, the duplication of resources and staffing might not provide a proportional return on investment. Additionally, there's a noticeable disconnect in the care continuum, from pre-hospital settings to primary and secondary care centers, district hospitals, private institutions, and tertiary care facilities. Duality of the challenges faced by emergency care in India requires coordinated participation of the government, public health facilities, private providers, healthcare institutions, civil society and sectors outside health like transport, education and communication to develop a high-quality integrated model for emergency care. Moreover, most of the attempts to solve the above-mentioned challenges are system-focused rather than patient-focused thus leaving a huge gap in the delivery and effectiveness of care. Mitigating gaps by adopting innovative strategies, leveraging technology, and focusing on developing competent human resources is necessary for providing comprehensive emergency care.

The INDIAEMS study aims to implement an integrated emergency care system addressing diverse medical emergencies across five diverse districts and offers a scalable model with significant policy implications, potentially revolutionizing emergency care delivery in India. This model, rooted in the challenges and learnings from diverse geographical and socio-economic contexts, aligns with India's healthcare goals and contributes to global health discussions, particularly for low- and middle-income countries. Reflecting the study's alignment with Sustainable Development Goals and national health objectives underscores the criticality of integrated healthcare systems in improving health standards. The implementation research, interweaving quantitative and qualitative methodologies, is expected to yield insights that could influence future healthcare policies and emergency care strategies nationally and globally. Possible pitfalls we expect during the implementation process are community participation, public-private sector disconnect, developing a supply chain for emergency care over a short period of the project, multisectoral stakeholder involvement, and changing disease spectrum.

### Contributions to the literature

- Provides robust evidence on the feasibility, effectiveness, and scalability of a patient-centric integrated emergency care model across diverse geographical and socio-economic contexts

- The study aims to offers practical insights into the application of implementation science frameworks for developing and optimizing complex health system interventions

- Generates valuable data on baseline EMD burdens, health-seeking behaviors, and health system preparedness in selected Indian districts, contributing to a better understanding of the specific challenges and needs that integrated emergency care models must address to improve population coverage and outcomes.

- The project will demonstrates a replicable methodology for co-designing, implementing, and evaluating a comprehensive emergency care system

## Supporting information

**S1 Table. Formative Research Components, Stakeholders, Data Collection Techniques and Sample Size.**
(PDF)

**S2 Table. Determinant framework used for implementing high quality emergency care.**
(PDF)

**S3 Table. Tracer indicators for quality of emergency care system.**
(PDF)

**S4 Table. Process indicators.**
(PDF)

**S5 Table. Evaluation of India EMS Model.**
(PDF)

**S1 Fig. Project management organogram.**
(PDF)

## Acknowledgments

We thank the project review committee members, for their valuable advise in protocol writing.

 **\*India EMS Study Group (In alphabetical order of State)**

Vanita Suri, Postgraduate Institute of Medical Education and Research, Chandigarh; Apurvakumar Pandya, Indian Institute of Public Health, Gandhinagar; Amit Ganatra, Ankita Priyadarshini, Hardik Dave, Nimesh Malaviya, Nupoor Nagar, Prachi Patel, Ravish Kshatriya, Sanket Saraiya, Saxena A K, Shaily Surti, Parul University, Vadodara, Gujarat; Subroto Das, Lifeline foundation, Suman Rao, St. John's Medical College Hospital, Upendra Bhojani, Institute of Public Health, Bengaluru, Karnataka; Abhijit P Pakhare, Abhishek Goel, Amit Agarwal, Bhupeshwari Patel, Bhushan Shah, K Pushplata, Jai Prakash Sharma, Manal Khan, Rajnish Joshi, All India Institute of Medical Sciences Bhopal, Madhya Pradesh; Nidhi Sharma Chauhan, Saifee Hospital, Rajshree Katke, Grant Government Medical College, S.K.Singh, International Institute of Population Sciences, Mumbai,Sandeep Salvi, Chest Research and Training (CREST), Pune, Maharashtra; Ambuj Roy, Anand Krishnan, Aparna Sharma K, Maneesh Singhal, All India Institute of Medical Sciences,New Delhi, Bala Subramanium, National Health System Resource Centre, Kanika Vasudeva, Indian Council of Medical Research, L. Swasticharan, Nirman Bhawan, New Delhi; Anbusenthil G, Indira Gandhi Government General Hospital & Post Graduate Institute, Duraisamy R, Govindarajan S, National Health Mission, Puducherry, Murali R, Narayanan D, Directorate of Health & Family Welfare Services, Puducherry; Kavita Vasudevan P, Surendar R, Udayashankar C, Indira Gandhi Medical College & Research Institute, Thirumalai Sanker R, Community Health Center Karikalampakkam; Anish Keepanasseril MS, Manju Rajaram, Nanda Kishore Maroju, Rajeswari Aghoram, Sitanshu Sekhar Kar, Jawaharlal Institute of Postgraduate Medical Education & Research; Valsa Diana, Rajiv Gandhi Maternity and Child Government Hospital, Puducherry; Gurbhej Singh, Pinki Pargal, Tapasya Dhar, Christian Medical College; Bishav Mohan, Dayanand Medical College & Hospital, Ludhiana;

Sandeep Singh Gill, Department of Health & Family Welfare, Punjab, Ashu Gupta, State Non Communicable Diseases Cell, Punjab; Peter John Victor, Vineeth Thomas, Christian Medical College, Vellore; Thomas Alexander, Kovai Medical Centre and Hospital, Coimbatore, Tamil Nadu; Ramana Rao, Emergency Medicine Learning Centre and Research, Sarang Deo, Indian School of Business, Hyderabad, Telangana; S Harikrishnan, P N Sylaja, Sree Chitra Tirunal Institute for Medical Sciences and Technology, Thiruvananthapuram; Asha P Shetty, Binod Kumar Patro, Chita Ranjan Mohanty, Payel Roy, Priyamadhaba Behera, Sanjeev Kumar Bhoi, Saroj Kumar Sahoo, Satyabrata Guru, Sonu Hangma Subba, Swagata Tripathy, Sweta Singh, Tapas Kumar Som, Upendra Hansda, All India Institute of Medical Sciences, Bhuvaneswar; Subhendu Kumar Acharya, Indian Council of Medical Research- Regional Medical Research Centre; Debasish Pandit, Kaushik Mishra, Sri Jagannath Medical College & Hospital, Bhubaneswar, Odisha, Debasish Pandit, Kaushik Mishra, SJMCH Puri, Odisha; Nilakantha Mishra, Niranjan Mishra, Sachidananda Mohanty, Santosh Kumar Mishra, Department of Health & Family Welfare, Government of Odisha, Vishwajeet Kumar, Community Empowerment Lab, Lucknow, Uttar Pradesh.

## Author contributions

**Conceptualization:** Manu Ayyan S, Arvind Kumar Singh, Hemantkumar S. Patadia, Shreyas Patel, Saurabh Saigal, Jeyaraj Durai Pandian, Ankur Joshi.

**Funding acquisition:** Manu Ayyan S, Arvind Kumar Singh, Hemantkumar S. Patadia, Shreyas Patel, Saurabh Saigal, Jeyaraj Durai Pandian, Meenakshi Sharma.

**Methodology:** Manu Ayyan S, Arvind Kumar Singh, Hemantkumar S. Patadia, Shreyas Patel, Saurabh Saigal, Jeyaraj Durai Pandian, Thejus Varghese, Ankur Joshi, Meenakshi Sharma.

**Project administration:** Manu Ayyan S, Arvind Kumar Singh, Hemantkumar S. Patadia, Shreyas Patel, Jeyaraj Durai Pandian.

**Resources:** Arvind Kumar Singh.

**Writing – original draft:** Manu Ayyan S, Arvind Kumar Singh, Hemantkumar S. Patadia, Shreyas Patel, Saurabh Saigal, Jeyaraj Durai Pandian, Thejus Varghese, Ankur Joshi, Meenakshi Sharma.

**Writing – review & editing:** Manu Ayyan S, Hemantkumar S. Patadia, Shreyas Patel, Saurabh Saigal, Jeyaraj Durai Pandian, Thejus Varghese, Ankur Joshi, Meenakshi Sharma.

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
