## [Decision Letter · Decision Letter 0]

5 Aug 2025

PONE-D-25-35153Developing a high-quality patient-centric integrated model for emergency care system in selected districts of India: An implementation research protocol (INDIA-EMS Study)PLOS ONE

Dear Dr. Sharma,

Thank you for submitting your manuscript to PLOS ONE. After careful consideration, we feel that it has merit but does not fully meet PLOS ONE’s publication criteria as it currently stands. Therefore, we invite you to submit a revised version of the manuscript that addresses the points raised during the review process.

We look forward to receiving your revised manuscript.

Kind regards,

Hariom Kumar Solanki, M.D.

Academic Editor

PLOS ONE

1. You may seek permission from the original copyright holder of Figure(s) [#] to publish the content specifically under the CC BY 4.0 license. 

6.Please review your reference list to ensure that it is complete and correct. If you have cited papers that have been retracted, please include the rationale for doing so in the manuscript text, or remove these references and replace them with relevant current references. Any changes to the reference list should be mentioned in the rebuttal letter that accompanies your revised manuscript. If you need to cite a retracted article, indicate the article’s retracted status in the References list and also include a citation and full reference for the retraction notice.

Additional Editor Comments:

It is a well written study protocol with substantial implications of the findings once the study is completed. However, Please address the following comments of the reviewers:

In Methodology part of abstract, write "Odisha" instead of "Orissa".

In the Study Populations and Sites subsection of Material and Methods, write the correct full form of NPNCD, which is "National Programme for Prevention and Control of Non-Communicable Diseases".

Thank you

Reviewers' comments:

Reviewer's Responses to Questions

**Comments to the Author**

1. Does the manuscript provide a valid rationale for the proposed study, with clearly identified and justified research questions?

Reviewer #1: Yes

Reviewer #2: Yes

2. Is the protocol technically sound and planned in a manner that will lead to a meaningful outcome and allow testing the stated hypotheses?

Reviewer #1: Yes

Reviewer #2: Yes

3. Is the methodology feasible and described in sufficient detail to allow the work to be replicable?

Reviewer #1: Yes

Reviewer #2: Yes

4. Have the authors described where all data underlying the findings will be made available when the study is complete?

Reviewer #1: Yes

Reviewer #2: Yes

5. Is the manuscript presented in an intelligible fashion and written in standard English?

Reviewer #1: Yes

Reviewer #2: Yes

6. Review Comments to the Author

You may also provide optional suggestions and comments to authors that they might find helpful in planning their study.

Reviewer #1: This is a very well written proposal and aims for a significant step in implementation of integrated emergency services in India compatible not just with health system but also with local governments and other stakeholders. It seems well thought-out, though, challenges and bottlenecks at the implementation level, from ground to up-top, and with different stakeholders, will need to be evaluated and managed through different but specific mechanisms. It would be interesting to see the final results of this implementation research.

Reviewer #2: In Methodology part of abstract, write "Odisha" instead of "Orissa".

In Introduction- "India is committed to achieving Sustainable Development Goal 3" add "by 2030"

In the Study Populations and Sites subsection of Material and Methods, write the correct full form of NPNCD, which is "National Programme for Prevention and Control of Non-Communicable Diseases".

7. PLOS authors have the option to publish the peer review history of their article (what does this mean? ). If published, this will include your full peer review and any attached files.

**Do you want your identity to be public for this peer review?** For information about this choice, including consent withdrawal, please see our Privacy Policy .

Reviewer #1: **Yes: ** Dr Yash Alok

Reviewer #2: No

---

## [Author Response · Author response to Decision Letter 1]

10 Aug 2025

We sincerely thank you and the reviewers for the constructive feedback on our manuscript PONE-D-25-35153 titled "Developing a high-quality patient-centric integrated model for emergency care system in selected districts of India: An implementation research protocol (INDIA-EMS Study)". We have carefully considered all comments and have made the necessary revisions to enhance the clarity and quality of our work. As detailed below, we have addressed each reviewer comment with corresponding revisions/responses:

Comment 1: Please ensure that your manuscript meets PLOS ONE's style requirements, including those for file naming. The PLOS ONE style templates can be found at https://journals.plos.org/plosone/s/file?id=wjVg/PLOSOne_formatting_sample_main_body.pdf

and https://journals.plos.org/plosone/s/file?id=ba62/PLOSOne_formatting_sample_title_authors_affiliations.pdf

Response: We sincerely appreciate your guidance on aligning our manuscript with PLOS ONE's formatting standards. In compliance with the provided templates, we have checked our manuscript to meet the specified style requirements. Additionally, we have ensured that all files are named according to the prescribed conventions.

Comment 2: PLOS requires an ORCID iD for the corresponding author in Editorial Manager on papers submitted after December 6th, 2016. Please ensure that you have an ORCID iD and that it is validated in Editorial Manager. To do this, go to ‘Update my Information’ (in the upper left-hand corner of the main menu), and click on the Fetch/Validate link next to the ORCID field. This will take you to the ORCID site and allow you to create a new iD or authenticate a pre-existing iD in Editorial Manager.

Response: We have now successfully added and validated the ORCID iD for the corresponding author in the Editorial Manager system as instructed.

Comment 3: Please include your full ethics statement in the ‘Methods’ section of your manuscript file. In your statement, please include the full name of the IRB or ethics committee who approved or waived your study, as well as whether or not you obtained informed written or verbal consent. If consent was waived for your study, please include this information in your statement as well.

Response: Thank you for highlighting the need for a comprehensive ethics statement. We confirm that a full statement has now been included in the Methods section (Lines 505–514) of the revised manuscript. It reads: “Ethical approval was received from the Christian Medical College and Hospital Institutional Ethics committee (Reference Number: BMHR-IEC/Apprvl-IndiaEMS/23-12-503/Neuro), Jawaharlal Institute of Postgraduate Medical Education and Research Institutional Ethics Committee for Observational Studies (Reference Number: JIP/IEC-OS/2023/352), Parul University Institutional Ethics Committee for Human Research (Reference Number: PUIECHR/PIMSR/00/081734/6401), All India Institute of Medical Sciences, Bhopal, Institutional Human Ethics Committee, (Reference Number: IHEC-LOP/2023/EL0116), All India Institute of Medical Sciences, Bhubaneshwar, Institutional Human Ethics Committee (Reference Number: T/EMF/CM&FM/2023-24/104) with additional permissions from respective district health authorities.”

Regarding the consent, the study involves the implementation of standard care treatment workflows requiring no participant consent. However, blanket consent will be obtained from district health administration or facility managers in accessing registers and facility-level data. Prospective study participants will be inducted only upon obtaining written informed consent, with an additional provision for assent in the case of younger participants. All individual in-depth interviews and focus group discussions will be conducted in confidential settings. Any adverse events or protocol amendments will be promptly reported to the Institutional Ethics Committee, ensuring no protocol changes without written consent. The same is mentioned in the manuscript from lines number 515 to 523.

Comment 4: We note that Figure 1 in your submission contain [map/satellite] images which may be copyrighted. All PLOS content is published under the Creative Commons Attribution License (CC BY 4.0), which means that the manuscript, images, and Supporting Information files will be freely available online, and any third party is permitted to access, download, copy, distribute, and use these materials in any way, even commercially, with proper attribution. For these reasons, we cannot publish previously copyrighted maps or satellite images created using proprietary data, such as Google software (Google Maps, Street View, and Earth). For more information, see our copyright guidelines: http://journals.plos.org/plosone/s/licenses-and-copyright.

Response: We have received the necessary permission to publish under CC 4.0. We have edited the figure caption per instructions and uploaded the permission letter from the creator.

Comment 5: If the reviewer comments include a recommendation to cite specific previously published works, please review and evaluate these publications to determine whether they are relevant and should be cited. There is no requirement to cite these works unless the editor has indicated otherwise.

Response: We noted your guidance regarding requests for citing specific previously published works. As no such recommendation was provided in the reviewer feedback, there were consequently no new citations to consider or evaluate for inclusion.

Comment 6: Please review your reference list to ensure that it is complete and correct. If you have cited papers that have been retracted, please include the rationale for doing so in the manuscript text, or remove these references and replace them with relevant current references. Any changes to the reference list should be mentioned in the rebuttal letter that accompanies your revised manuscript. If you need to cite a retracted article, indicate the article’s retracted status in the References list and also include a citation and full reference for the retraction notice.

Response: Thank you for your direction regarding the reference list. We have carefully reviewed our citations and confirm that no changes were needed. Our reference list remains complete, accurate, and free from retracted articles in accordance with journal guidelines.

Comment 7: It is a well-written study protocol with substantial implications of the findings once the study is completed. However, Please address the following comments of the reviewers:

In Methodology part of abstract, write "Odisha" instead of "Orissa".

Response: We have updated the Methodology section of the abstract, replacing “Orissa” with the correct and current name, “Odisha”, to reflect the appropriate nomenclature (Line no.36).

Comment 8: In the Study Populations and Sites subsection of Material and Methods, write the correct full form of NPNCD, which is "National Programme for Prevention and Control of Non-Communicable Diseases".

Response: We have corrected the full form of NPNCD to “National Programme for Prevention and Control of Non-Communicable Diseases” in the Study Populations and Sites subsection of the Methods section (Lines no. 156-157).

Comment 9: In Introduction- "India is committed to achieving Sustainable Development Goal 3" add "by 2030"

Response: Thank you for the suggestion. We have updated the Introduction sentence (Line no.80).

We believe that the above revisions/responses have addressed the reviewers' concerns and have strengthened the manuscript. We are grateful for the reviewers' thoughtful suggestions and are confident that the revised version is now suitable for publication. We look forward to your feedback and hope that the manuscript will meet the expectations of the editorial team and reviewers.

Thank you once again for your time and consideration.

---

## [Editor Report · Decision Letter 1]

14 Aug 2025

Developing a high-quality patient-centric integrated model for emergency care system in selected districts of India: An implementation research protocol (INDIA-EMS Study)

PONE-D-25-35153R1

Dear Dr. Sharma,

We’re pleased to inform you that your manuscript has been judged scientifically suitable for publication and will be formally accepted for publication once it meets all outstanding technical requirements.

Kind regards,

Hariom Kumar Solanki, M.D.

Academic Editor

PLOS ONE
---

## [Editor Report · Acceptance letter]

PONE-D-25-35153R1

PLOS ONE

Dear Dr. Sharma,

I'm pleased to inform you that your manuscript has been deemed suitable for publication in PLOS ONE. Congratulations! Your manuscript is now being handed over to our production team.

Kind regards,

on behalf of

Dr. Hariom Kumar Solanki

Academic Editor

PLOS ONE